# Deep-Learning-Based Segmentation of the Shoulder from MRI with Inference Accuracy Prediction

**DOI:** 10.3390/diagnostics13101668

**Published:** 2023-05-09

**Authors:** Hanspeter Hess, Adrian C. Ruckli, Finn Bürki, Nicolas Gerber, Jennifer Menzemer, Jürgen Burger, Michael Schär, Matthias A. Zumstein, Kate Gerber

**Affiliations:** 1School of Biomedical and Precision Engineering, Personalised Medicine Research, University of Bern, 3008 Bern, Switzerland; hanspeter.hess@unibe.ch (H.H.); adrian.ruckli@unibe.ch (A.C.R.); finn.buerki@students.unibe.ch (F.B.); nicolas.gerber@unibe.ch (N.G.); juergen.burger@med.unibe.ch (J.B.); 2Shoulder, Elbow and Orthopaedic Sports Medicine, Orthopädie Sonnenhof, 3006 Bern, Switzerland; jennifer.menzemer@gmail.com (J.M.); matthiaszumstein@sonnenhof.ch (M.A.Z.); 3Department of Orthopaedic Surgery and Traumatology, Inselspital, University Hospital of Bern, 3010 Bern, Switzerland; michael.schaer@insel.ch

**Keywords:** shoulder, rotator cuff, deep learning, MRI, automatic segmentation, segmentation accuracy prediction

## Abstract

Three-dimensional (3D)-image-based anatomical analysis of rotator cuff tear patients has been proposed as a way to improve repair prognosis analysis to reduce the incidence of postoperative retear. However, for application in clinics, an efficient and robust method for the segmentation of anatomy from MRI is required. We present the use of a deep learning network for automatic segmentation of the humerus, scapula, and rotator cuff muscles with integrated automatic result verification. Trained on N = 111 and tested on N = 60 diagnostic T1-weighted MRI of 76 rotator cuff tear patients acquired from 19 centers, a nnU-Net segmented the anatomy with an average Dice coefficient of 0.91 ± 0.06. For the automatic identification of inaccurate segmentations during the inference procedure, the nnU-Net framework was adapted to allow for the estimation of label-specific network uncertainty directly from its subnetworks. The average Dice coefficient of segmentation results from the subnetworks identified labels requiring segmentation correction with an average sensitivity of 1.0 and a specificity of 0.94. The presented automatic methods facilitate the use of 3D diagnosis in clinical routine by eliminating the need for time-consuming manual segmentation and slice-by-slice segmentation verification.

## 1. Introduction

Acute or chronic rotator cuff tendon tears (RCTs) affect almost 10% of the general adult population [1] and are associated with pain and decreased shoulder function [2]. After the failure of conservative treatment, surgical rotator cuff repair (RCR), in which the torn tendon is reattached to the bone via sutures and/or bone anchors, is the preferred treatment option. Repair is associated with significant short- and long-term improvements in pain, function, and strength [3,4]. However, not all patients benefit from a repair procedure. The retearing of the tendon following repair presents a considerable problem, affecting 20–50% of all repairs [3,5]. In cases of structural and clinical failure, an additional more invasive treatment alternative such as tendon transfer or reverse shoulder arthroplasty may be indicated.

For the preoperative identification of patients predisposed to RCR failure, several image-based diagnostic factors parameterizing the bony morphology and tissue quality of the rotator cuff muscles and tendons have been identified as predictors for repair outcome success [6,7,8,9]. To enable measurement in clinical routine, these factors are assessed on single slices of RCT diagnostic magnetic resonance images (MRIs) [10] or on supplementary radiographs [8,9]. However, two-dimensional (2D) analysis of the anatomy on selected representative slices may be misrepresentative of the overall anatomy, and analyses performed on radiographs may suffer from projection error [11,12]. More holistic three-dimensional (3D) anatomical analysis, utilizing 3D models from the segmentation of computed tomography (CT) or MRI, has been performed in smaller studies [13,14,15], but the requirement of time-consuming manual slice-by-slice segmentation of the anatomy in the image data has prevented any wider application.

To improve the efficiency and accuracy of anatomical analysis of the rotator cuff, the use of deep-learning-based algorithms for the automatic segmentation of anatomical structures from 3D image data has been described. Medina et al. and Ro et al. reported the use of U-net, a deep learning network architecture for semantic segmentation [16], to automatically segment the rotator cuff muscles on a single 2D slice (Y-view) of T1-weighted MRI for automatic fat fraction analysis [17,18]. In 3D, Godoy et al. reported the use of U-net for automatic segmentation of the pectoralis major muscle on T1-weighted MRI to automatically evaluate its major cross-sectional area [19]. Most recently, Riem et al. utilized U-net for automatic segmentation and 3D fat fraction analysis of the rotator cuff muscles from clinical diagnostic sagittal T1-weighted MRI scans [20]. They reported mean and standard deviation segmentation accuracy of 0.92 ± 0.14; however, the method was trained and tested on data from only two centers, and only segmentation of the sagittal MRI was performed, inhibiting the analysis of the muscle volume and the fat fraction in the entire muscle due to the restricted field of view (FOV). The 3D automatic segmentation of MRI images presents challenges related to high variances in image orientation, resolution, and signal intensity and contrast due to the use of different coils, magnetic field strengths, and vendor-specific image processing techniques. The intraarticular injection of contrast agent for RCT diagnosis [21] further adds to image intensity variance. For the general application of a segmentation network in the analysis of the rotator cuff in multicenter clinical trials or clinical diagnosis, a model that can exploit the information from all available MRI views and which is validated on heterogeneous data acquired from multiple centers is required.

However, the verification of a segmentation network on all possible variances in anatomy and image quality is infeasible, and thus, even in the case of high accuracy and robustness, deep learning networks can generate anatomy segmentations with insufficient accuracy for effective clinical analysis. This uncertainty in network performance results in the need for expert manual verification of each segmentation result, slice by slice, during application.

To reduce the burden of manual verification, methods for the automatic estimation of segmentation accuracy based on the epistemic uncertainty of deep learning segmentation networks have recently been proposed. Roy et al. correlated structure-wise aggregated uncertainties with the overlap between the automatic and manual ground-truth segmentation [22]. The uncertainty estimate was based on comparisons of multiple segmentations generated by the application of random dropout during network test times. More recently, Jungo et al. achieved improved results from segmentation uncertainty calculated from segmentations generated from auxiliary networks trained with different subsets of training samples [23,24]. However, the advantage of combining these two methods via the application of dropout on the auxiliary trained networks has not yet been studied, and the efficiency of these methods applied to the segmentation of musculoskeletal structures in MRI has yet to be reported.

First introduced by Isensee et al. in 2021, nnU-Net, a fully automated framework for the semantic segmentation of biomedical images, has been gaining popularity due to reported high-accuracy segmentation in various tasks [25] without the need for task-specific tuning. The nnU-Net framework maximizes segmentation accuracy during test time via the ensemble of results of multiple, differently trained networks. We hypothesize that the variance in the predictions of the nnU-Net subnetworks can be used for the automatic detection of inaccurate segmentations directly, eliminating the need for additional auxiliary networks for automatic accuracy verification.

To enable widespread accurate and efficient 3D measurements of RCT diagnostic factors, we propose a fully automatic algorithm for 3D segmentation of the shoulder joint from diagnostic MRI with integrated automatic segmentation error checking. We present the use of nnU-Net for shoulder anatomy segmentation from diagnostic MRI from all planes (axial, sagittal, and transversal), verified on heterogeneous data acquired in multiple centers. For efficient accuracy verification, we present a study of automatic uncertainty-based label-specific error checking, utilizing multiple segmentations generated from both the nnU-Net subnetworks and dropout during inference, integrated directly into the nnU-net framework.

## 2. Materials and Methods

Networks for the automatic segmentation of rotator cuff anatomy and for automatic segmentation error detection were trained on routine RCT diagnostic MRI and verified on unseen data acquired from multiple centers. This study was conducted in accordance with the Declaration of Helsinki and approved by the Institutional Review Board of the ethical commission of Bern (no.: 2021-00326).

### 2.1. Anatomy Segmentation

#### 2.1.1. Data

Retrospective MR arthrography data (N = 171) from patients with posterosuperior rotator cuff tears seen between 2017 and 2021 at the Orthopedic outpatient clinic of Sonnenhof, Bern, Switzerland, were collected and coded. Despite all patients being seen at the same clinic, the MRI data were acquired from 9 different institutions during routine diagnosis, resulting in a significant degree of variability in slice thicknesses and in-plane resolution (Table 1). The data were randomly divided into training and test datasets. The training dataset consisted of N = 111, T1-weighted MRI of 37 shoulders, obtained from 7 institutions, and acquired in coronal (N = 37), sagittal (N = 37), and transversal (N = 37) orientations (Table 1). The test dataset consisted of N = 60, T1-weighted MRI of 39 shoulders, obtained from 9 institutions, and acquired in coronal (N = 20), sagittal (N = 20), and transversal (N = 20) orientations (Table 1 and Figure 1).

Manual semantic segmentation of the rotator cuff anatomy, including the humerus, scapula, and rotator cuff muscles—supraspinatus (SSP), infraspinatus (ISP), subscapularis (SSC), and teres minor (TM)—was performed by an expert in the complete FOV of the MRI data.

#### 2.1.2. Segmentation Network Training

A nnU-Net was trained for automatic segmentation of the shoulder anatomy. The automatic internal procedures provided by nnU-Net, such as data fingerprinting for a data-specific network setup, pre-processing techniques, and automatic 5-fold cross validation procedure, were applied. The 2D and 3D networks were trained for 150 epochs on all MRI orientations together. During the training process, the label-wise accuracy of the automatic segmentation of the 2D and 3D networks as well as the ensemble result of these networks were calculated as the mean Dice coefficient relative to the manual ground truth.

#### 2.1.3. Segmentation Network Verification

The segmentation accuracy of the best performing network was verified on the unseen test data. Test time augmentation (*TTA*) and ensemble of the results from the different networks trained during 5-fold cross validation training procedure were applied to maximize the network performance. The Dice coefficient between automatic segmentation and manual ground truth was calculated for each label.

### 2.2. Inference Accuracy Prediction

For automatic prediction of the segmentation accuracy, network uncertainty was estimated as the agreement of auxiliary segmentations. The performance of various uncertainty metrics in predicting segmentation accuracy from a varying number of auxiliary segmentations was investigated and used to calibrate thresholds of the uncertainty metrics. The sensitivity and specificity of the best performing method in detecting segmentations with insufficient accuracy, generated by the previously described segmentation network, were then determined.

#### 2.2.1. Auxiliary Predictions for Uncertainty Calculation

Auxiliary segmentations were generated from the nnU-Net subnetworks and from the application of dropout during the inference process. The nnU-Net framework was augmented to generate the segmentation prediction (*P_n_*) of each of the N = 5 subnetworks trained during the 5-fold cross validation process by applying the argmax operator to the softmax output (Figure 1). In addition, a Bayesian neural network was approximated to generate stochastic network samples by applying test time dropout (*TTD*) [26] to each of the five subnetworks. The nnU-Net framework was modified to allow activation of dropout layers during the inference process, thereby allowing for the generation of a greater number (*N* = 10, 15, 20) of auxiliary inference segmentations (*P_n_*). *TTD* probabilities (*p =* 0.05, 0.15, 0.25) were applied: (a) to the whole network and (b) only to the encoder path, including the network’s bottleneck.

#### 2.2.2. Accuracy Prediction Metrics

To estimate the label-specific prediction accuracy of each label (*l*) of a given prediction (*P*), the disagreement of the auxiliary subnetwork predictions (*P_n_*) was measured using seven different metrics: (1) intersection over union of the auxiliary predictions (*IoU_l_*); (2) intersection over the final inference prediction (*IoP_l_*); (3) the *IoU_l_* and (4) the *IoP_l_* represented as Dice coefficients; (5) the average (*DCA_l_*) and (6) the median (*DCM_l_*) of the Dice coefficients calculated between each of the auxiliary predictions and the final prediction; and (7) the coefficient of variation in the label volume (*CV_l_*) between all auxiliary predictions
(1)$$IoU_{l}\;=\;\frac{\left|\left(P_{1}=l\right)\;\cap \;\left(P_{2}=l\right)\;\cap \;\ldots \;\cap \;\left(P_{n}=l\right)\right|}{\left|\left(P_{1}=l\right)\;\cup \;\left(P_{2}=l\right)\;\cup \;\ldots \cup \;\left(P_{n}=l\right)\right|}$$
(2)$$IoP_{l}=\frac{\left|\left(P_{1}=l\right)\;\cap \;\left(P_{2}=l\right)\;\cap \;\ldots \;\cap \;\left(P_{n}=l\right)\right|}{\left|\left(P=l\right)\right|}$$
(3)$$DC_{IoU_{l}}\;=\;\frac{2IoU_{l}}{1+IoU_{l}}$$
(4)$$DC_{IoP_{l}}\;=\;\frac{2IoP_{l}}{1+IoP_{l}}$$
(5)$$DCA_{l}\;=\;Mean\left(Dice\left(P_{i}=l,P=l\right)\right)$$
(6)$$DCM_{l}\;=\;Median\left(Dice\left(P_{i}=l,P=l\right)\right)$$
(7)$$CV_{l}=\frac{\sigma_{l}}{\mu_{l}},\;\mathrm{with}\;\mathrm{mean}\;(\mu_{l})\;\mathrm{and}\;\mathrm{standard}\;\mathrm{deviation}\;(\sigma_{l})\;\mathrm{of}\;\mathrm{the}\;\mathrm{label}\;\mathrm{volumes}.$$

#### 2.2.3. Inference Accuracy Prediction Performance

To evaluate the performance of the inference accuracy prediction using the described metrics and to calibrate thresholds for the detection of inaccurate segmentations, an additional, less robust calibration network was trained to intentionally increase the variation in segmentation accuracies (Figure 1a). The 3D nnU-Net was trained for 150 epochs on a randomly chosen subset of N = 60 MRIs (20 shoulders) from the training set (coronal, N = 20; sagittal, N = 20; transversal, N = 20 (Figure 2)). The trained network was then applied without post-processing to the remaining 51 MRIs (17 shoulders; coronal, N = 17; sagittal, N = 17; transversal, N = 17) to generate inference segmentations. These segmentations were visually inspected and manually corrected, if required, by an expert. The label-specific extent of manual correction was quantified as the Dice coefficient between the raw prediction result and corrected segmentation for each label. The extent of correction was used as the measure for inference segmentation accuracy (rather than a comparison to the independent manual ground-truth segmentation) as it better represents the results of the manual correction process that occurs during practical application of segmentation networks. The performance of each accuracy prediction metric was evaluated as the coefficient of determination (R^2^) between the metric value and the extent of required segmentation for each label (*l*) for each of the combinations of auxiliary subnetwork predictions. The five best performing metrics and the corresponding auxiliary subnetwork prediction methods were identified.

#### 2.2.4. Segmentation Goodness Classification

In this study, the segmentation goodness classification was defined by considering the extent of necessary manual correction of the final ensemble prediction. Thresholds on the Dice coefficient between the raw prediction result and corrected segmentation were defined as 97.5% for bones and 92.5% for muscles. To identify the optimal label-specific thresholds on the prediction metric values for automatic segmentation goodness classification, the general Youden index [27] was calculated with a greater weight on sensitivity (relative loss = 2) for the top-five performing prediction metrics and the corresponding auxiliary subnetwork prediction methods. The sensitivities and specificities of the five predictive measures for categorizing inaccurate and accurate segmentations were calculated.

#### 2.2.5. Performance Testing of Goodness Classification

The performance of the best performing auxiliary subnetwork prediction method and segmentation goodness categorization thresholds found with the calibration network were tested on the 3D segmentation network trained with the complete training dataset (Figure 1b). To assess the performance of the error detection method on data with large variance, additional N = 3 fat-suppressed T1-weighted MRI (T1-fs-MRI) along all directions (coronal, N = 1; sagittal, N = 1; transversal, N = 1) were added to the test dataset, resulting in a complete performance test dataset of N = 63 image volumes. Inference results were manually corrected (if required, as determined by an expert clinician) and classified as insufficiently accurate if the extent of manual correction exceeded the thresholds previously defined during calibration. The performances of the automatic classification of segmentation goodness for each label were compared to the classification defined by the extent of manual correction.

## 3. Results

### 3.1. Anatomy Segmentation

The performances of the 2D and 3D ensemble networks during five-fold cross validation training are given in Table 2. Overall, the ensemble of the 2D and 3D networks did not improve the performance of the network compared to the 3D network.

On the unseen test dataset, the 3D network achieved mean and standard deviation segmentation accuracies of humerus, 0.992 ± 0.01; scapula, 0.978 ± 0.01; SSP, 0.981 ± 0.01; SSC, 0.974 ± 0.02; ISP, 0.974 ± 0.02; and TM, 0.94 ± 0.11. The automatic 3D segmentation result overlaid on the input MRI of a case with average segmentation accuracy is shown in Figure 3.

### 3.2. Inference Accuracy Prediction

The coefficient of determination (R^2^) of each uncertainty metric applied to the auxiliary segmentations generated from the subnetworks with different dropout settings, in predicting the segmentation accuracy of the 3D calibration network, is depicted in Figure 4. In general, higher dropout probabilities (*p*) decreased the metric performance, as well as dropout activated in the whole network compared to activation only in the encoder and bottleneck. Overall, the best performances were achieved using the individual predictions (*P_n_*) from the five subnetworks without dropout. In general, *TTA* slightly increased the performance, especially for networks where dropout was applied, with, however, an inference deceleration of a factor of eight. The *DCM* with *TTA* achieved the highest R^2^ value of 0.61, followed by the *DCA* with and without *TTA,* with R^2^ values of 0.60. The *DCA* with the lowest dropout probability of 0.05, *N* = 10 and 15 predictions, with *TTD* in the encoder and bottleneck and with *TTA* achieved the next best results with an R^2^ value of 0.57.

All of the top-five subnetwork and metric combinations detected insufficiently accurate inference segmentations, with a sensitivity of 1.0. The *DCA* without *TTA* and without *TTD* achieved the highest average specificity of 0.83 (Table 3). 

### 3.3. Performance Testing of Goodness Classification

Due to the overall high performance of the segmentation network, only four (1.1%) labels in the T1-weighted MRI in the test dataset were segmented with insufficient accuracy. Including the T1-fat-suppressed weighted MRI, a total of 4.7% of the labels were segmented with insufficient accuracy. As shown in Table 4, the *DCA* metric and the defined label-specific thresholds allowed all cases with insufficient accuracy to be detected with an overall specificity of 0.94.

The insufficiently accurate segmentations on the T1-weighted MRI originated from two patients (Figure 5). In the coronal and sagittal MRI of a patient with a TM with high fatty infiltration (Pat 1 in Figure 5), the algorithm failed in segmenting the TM accurately. The sagittal MRI of another patient (Pat 2 in Figure 5) showed prominent image artifacts, leading to failed segmentations of the SSC, the ISP, and the TM. The fat-suppressed T1-weighted MRI along all planes showed poor segmentations of the bones, SSC, and ISP (Pat 3 in Figure 5).

## 4. Discussion

Despite significant research on the causes of rotator cuff repair failure and the clinical use of prognostic factors for patient selection, failure rates remain high. Research on better predictive models is ongoing but still limited by the variability in and inaccuracy of manually performed 2D-image-based measurements [28]. Patient-specific 3D models of the shoulder would allow for automatic 3D analysis of rotator cuff reparability, the possible identification of novel 3D predictive factors, and the inclusion of 3D metrics in predictive model analysis. In this work, we present, for the first time, a network for fully automatic 3D semantic segmentation of bony structures (humerus and scapula) as well as the rotator cuff muscles on T1-weighted MRI along all planes from rotator cuff tear patients. The presented segmentation network demonstrated high accuracy on anisotropic diagnostic MRI images from different centers with a high variety of slice thicknesses and in-plane resolutions. The accuracy of the automatic bone segmentation was found to be comparable to other reports of 3D automatic bone segmentation from MRI, such as the use of a convolutional network for the segmentation of the pelvis and femur presented by Zeng et al. [29]. On the rotator cuff muscles, the segmentation network presented herein, when applied to multicenter data of all image planes, achieved similar segmentation accuracies to those reported by Riem et al. [20].

In this work, to achieve automatic error detection without additional expensive calculations, we augmented the well-established nnU-Net framework with the integration of a label-wise method for detecting inference segmentations with insufficient accuracy. From a large range of inference segmentation accuracy prediction metrics, the average Dice coefficient (*DCA*) of the segmentations from the differently trained networks without using *TTA* demonstrated the best performance with an average sensitivity and specificity of 1.0 and 0.94, respectively. This result adds further evidence to the finding of Jungo et al. [24] that estimating the network uncertainty by comparison of segmentations generated by differently trained networks achieves the best results overall. The application of *TTA* or *TTD* did not further improve the precision of the inference accuracy predictors.

Overall, the coefficients of determination of the accuracy prediction metrics and the Dice coefficient presented herein are lower compared to those presented by Roy et al. [22]. However, in this work, the values of the inference performance prediction metrics were compared to the Dice coefficients between the predicted and the manually corrected inference segmentation rather than the Dice coefficients between the predicted and original manual segmentation. Consequently, in this work, the maximal Dice coefficient of 1.0, which represented any segmentation that was completely accepted by the expert without the need for correction, was sometimes achieved. While we believe this measure is more representative of the state-of-the-art manual error verification process applied in clinical practice, it results in nonuniformly distributed Dice coefficients and lowers the coefficients of determination between the Dice and the presented inference performance predictors.

The goodness classification of the inference segmentation is a further crucial step toward the application of such inference performance prediction algorithms to large datasets and large clinical trials, as well as for the successful integration of methods into software for clinical diagnosis. In this work, inference segmentations with insufficient accuracy were defined as those exceeding a certain extent of manual correction (the amount of disagreement in the automatic segmentation results by an expert user). Ideally, accuracy thresholds should consider the sensitivity of the 3D clinical metric to the segmentation accuracy and the required final clinical metric precision. In this study, a higher accuracy acceptance threshold was selected for bone than for muscle for two reasons: (1) previous reports have suggested that the network would perform with slightly higher accuracy on bones than on muscle and (2) clinical measures of bone morphology would require landmark detection on the 3D bone surface and, thus, may be more sensitive to segmentation error than measures performed on the entire muscle, such as muscle volume and fat fraction measures. In the future, we recommend that accuracy acceptance thresholds alternatively be defined based on the studied effect of the final clinical metric calculated based on the segmentation. Additionally, location-specific uncertainty measurements could be used to provide precise performance predictions to eliminate unnecessary time spent correcting areas insignificant for the calculation of the final clinical metric.

With the use of a high-accuracy segmentation algorithm, such as the one presented herein, the majority of cases can be segmented completely automatically without manual correction or intervention. Inaccurate segmentations are accurately detected via the proposed automatic error-checking algorithm, providing users with high confidence in the use of the resulting 3D anatomical models for further clinical analysis while reducing labor-intensive, manual slice-by-slice monitoring of each inference result. Cases detected as insufficiently accurate would currently still require manual slice-by-slice verification and voxel-wise correction. In future work, pixel-wise or regional uncertainty metrics could be used to develop semiautomatic correction techniques [30]. Corrections based on the results of auxiliary networks could be suggested to the user, reducing the need for voxel-wise correction.

The algorithms presented herein were designed and tested on standard T1-weighted MRI sequences, which are commonly acquired for the diagnosis of rotator cuff tears. When applied to MRI sequences underrepresented in the training dataset, the network was more likely to segment the muscles with insufficient accuracy. While these erroneous predictions were recognized by the proposed error detection algorithm, we recommend also training networks with other MRI sequences, such as proton-density- or T2-weighted MRI, if segmentation of the shoulder structures from these MRI sequences is required.

The proposed automatic error-checking methods could additionally be applied in the automatic segmentation of other anatomical structures; however, for optimized performance on these structures, we recommend the re-evaluation of the prediction metrics and the recalibration of label-specific thresholds. For use in further research, the code of the augmented nnU-Net used within this study was made publicly available on GitHub.

In the future, the proposed shoulder anatomy segmentation methods will be used as the basis for the 3D analysis of the anatomy of rotator cuff patients. For example, in a multicenter study of arthroscopic rotator cuff repair [28], the algorithms have been applied to study the predictive value of 3D-image-based anatomical metrics, such as whole muscle and regional fat fraction, muscle volume, and 3D bone morphology metrics, for repair outcome. For use in future research, the presented segmentation network with integrated error detection will be made available upon request to the corresponding author.

## 5. Conclusions

The automatic and accurate segmentation of the shoulder joint in MRI of rotator cuff tear patients has the potential to allow efficient 3D patient-specific anatomy analysis for improved rotator cuff repair prognosis evaluation. In this work, the first fully automatic and accurate deep learning algorithm for semantic segmentation of the humerus, scapula, and rotator cuff muscles on multicenter MRI with automatic error detection was presented. It is hoped that this implementation will enable the widespread use of automatic 3D rotator cuff analysis, thereby eliminating errors associated with 2D measures and allowing for a more accurate and holistic patient-specific analysis of the rotator cuff anatomy for treatment optimization.

## Figures and Tables

**Figure 1 diagnostics-13-01668-f001:**
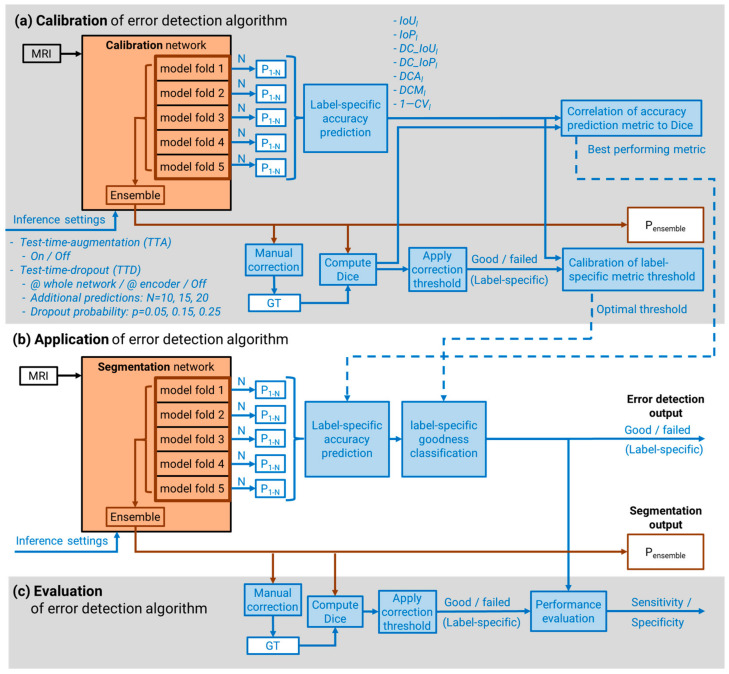
For the calibration of the error detection algorithm (**a**), different inference settings were applied on the calibration network during test time. The original nnU-Net (highlighted in red) was augmented (highlighted in blue) to generate inference segmentations (P_1-N_) from each subnetwork during test time. Predictions with and without test time augmentation (*TTA*) were tested. The best accuracy prediction metric was evaluated by analyzing the correlation between the metric values and the Dice coefficient between ground-truth (GT) and ensemble inference results (P_ensemble_). Inference segmentations requiring extensive manual correction (Dice falls below the correction threshold) were categorized as failed. The optimal label-specific thresholds of the accuracy prediction metric for automatic goodness classification were calibrated. During application of the error detection algorithm (**b**), the best accuracy prediction metric and label-specific thresholds are applied to the segmentation network during test time. For evaluation of the algorithm, the sensitivity and specificity of the automatically generated error detection were calculated (**c**).

**Figure 2 diagnostics-13-01668-f002:**
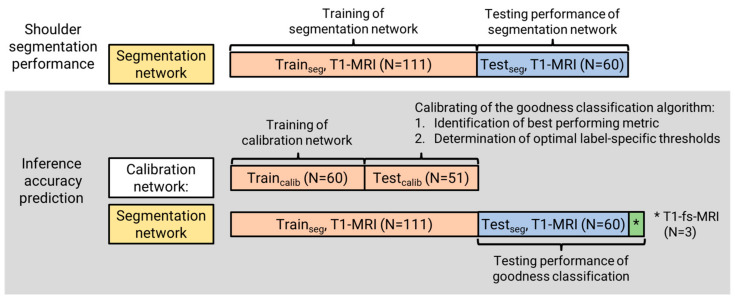
The training and test datasets were used to train and test the segmentation network. A subset of the training dataset was used to train the calibration network, while the remaining data were used to find the best performing inference accuracy prediction metric and the optimal threshold for goodness classification. The test dataset plus three additional T1-fs-MRIs were used to test the goodness classification when applied to the output of the segmentation network.

**Figure 3 diagnostics-13-01668-f003:**
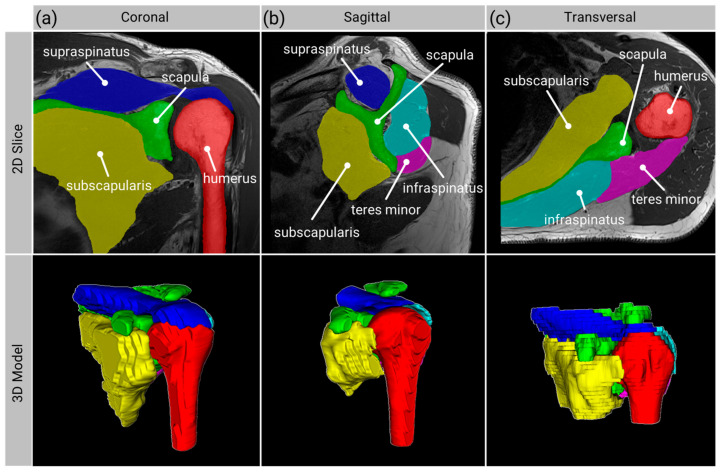
T1-MRI along all planes from the same patient, on which average segmentation accuracy was achieved. Top row: T1-weighted MRI with overlaid automatic segmentation results. Bottom row: 3D visualization of the automatic segmentation. (**a**) coronal plane, (**b**) sagittal plane, (**c**) transversal plane.

**Figure 4 diagnostics-13-01668-f004:**
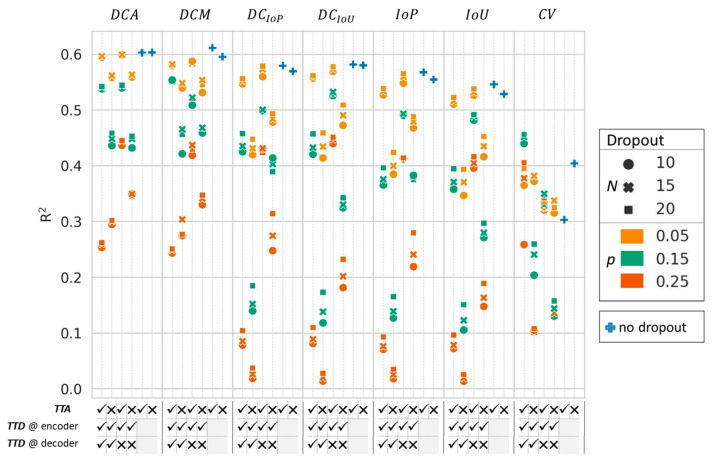
Coefficient of determination (R^2^) for the different inference accuracy prediction metrics applied to 51 segmentations. *N*—number of predictions with dropout, *p*—dropout probability, *TTA*—test time augmentation, *TTD*—test time dropout, *DCA*—average Dice, *DCM*—median Dice, DC*_IoP_*—*IoP* represented as Dice, DC*_IoU_*—*IoU* represented as Dice, *IoP*—intersection over prediction, *IoU*—intersection over union, *CV*—coefficient of variation.

**Figure 5 diagnostics-13-01668-f005:**
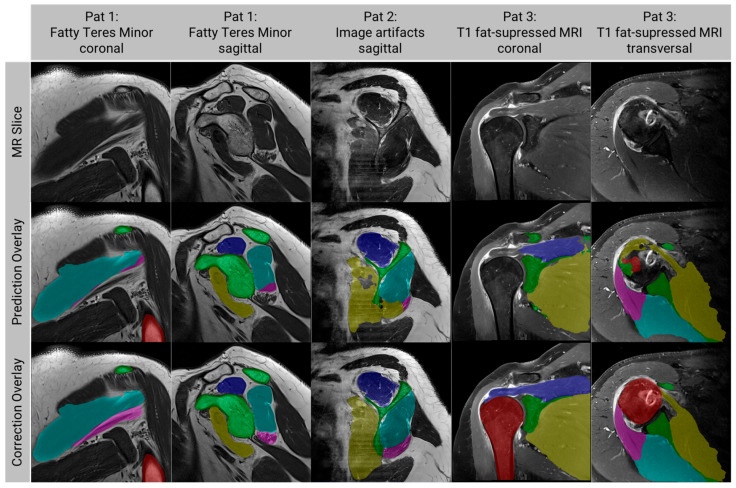
Examples of insufficiently accurate segmentations on the T1-weighted MRI. Top row: original MRI, middle row: overlay of the MRI and the automatic segmentation, bottom row: overlay of the MRI and the corrected segmentation. Humerus (red), scapula (green), supraspinatus (dark blue), subscapularis (yellow), infraspinatus (cyan), teres minor (magenta).

**Table 1 diagnostics-13-01668-t001:** Demographic and MRI data characteristics of the training and test datasets.

Characteristic	Training Dataset	Test Dataset
Image datasets, no.	111 T1-MRI	60 T1-MRI
Shoulders (Patients), no.	37 (37)	39 (38)
Age (years; mean ± std)	57 ± 13	57 ± 10
Female sex, no. (%)	16 (43)	13 (33)
Left side, no. (%)	15 (41)	14 (36)
Anatomical orientation, no.	37 coronal 37 sagittal 37 transversal	20 coronal 20 sagittal 20 transversal
Institutions, no.	7	9
Vendor no. (%)	35 (95) Siemens 2 (5) Philipps	32 (82) Siemens 7 (18) Philipps
Magnetic field strength no. (%)	21 (57) 1.5 Tesla 16 (43) 3.0 Tesla	14 (36) 1.5 Tesla 25 (64) 3.0 Tesla
In-plane resolution (mm)	0.2–0.5	0.2–0.5
Slice thickness (mm)	2.5–4.0	2.5–4.0
Spacing between slices (mm)	3.3–4.6	2.75–4.8

**Table 2 diagnostics-13-01668-t002:** Automatic shoulder segmentation accuracy per structure in Dice coefficient (mean ± standard deviation) compared to manual segmentation, for the 2D and 3D network and the ensemble of these networks for the humerus, scapula, supraspinatus (SSP), subscapularis (SSC), infraspinatus (ISP), teres minor (TM).

Network	Humerus	Scapula	SSP	SSC	ISP	TM	Overall
2D	0.96 ± 0.03	0.91 ± 0.05	0.90 ± 0.06	0.90 ± 0.09	0.89 ± 0.06	0.83 ± 0.09	0.90 ± 0.08
3D	0.97 ± 0.02	0.92 ± 0.05	0.92 ± 0.06	0.91 ± 0.07	0.91 ± 0.05	0.86 ± 0.07	0.91 ± 0.06
Ensemble	0.96 ± 0.02	0.92 ± 0.05	0.91 ± 0.05	0.91 ± 0.08	0.91 ± 0.05	0.85 ± 0.08	0.91 ± 0.07

**Table 3 diagnostics-13-01668-t003:** Achieved specificity of detecting insufficiently accurate segmentation by the top-five inference accuracy prediction metrics. All metrics achieved a sensitivity of 1.0.

Label	*DCM* *TTA*	*DCA* no *TTA*	*DCA* *TTA*	*DCA TTA* *N* = 10, *p* = 0.05	*DCA TTA* *N* = 15, *p* = 0.05
Humerus	0.98	1.00	1.00	1.00	1.00
Scapula	0.41	0.76	0.57	0.54	0.54
Supraspinatus	0.91	0.93	0.98	0.98	0.98
Subscapularis	0.92	0.94	0.94	0.96	0.98
Infraspinatus	0.84	0.84	0.84	0.84	0.82
Teres minor	0.59	0.50	0.53	0.50	0.53
Average	0.77	0.83	0.81	0.80	0.81

**Table 4 diagnostics-13-01668-t004:** Performance of the *DCA* metric without *TTA* without *TTD* with label-specific threshold to detect insufficiently accurate segmentations on the T1- and T1-fs-weighted MRI of the test set.

Label	Sensitivity	Specificity	TP	TN	FP	FN	Threshold
Humerus	1.00	1.00	3	60	0	0	0.97
Scapula	1.00	0.98	3	59	1	0	0.94
Supraspinatus	1.00	0.97	2	59	2	0	0.92
Subscapularis	1.00	0.97	3	58	2	0	0.91
Infraspinatus	1.00	0.98	4	58	1	0	0.93
Teres Minor	1.00	0.72	3	43	17	0	0.94

## Data Availability

The models and algorithms presented in this manuscript, along with the complete results are available upon request to the corresponding author. The code of the adapted nnU-Net was made publicly available on GitHub (https://github.com/persmed/nnUNet_structure_wise_uncertainty (accessed on 1 May 2023)).

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
