# Peer review of "Deep-Learning-Based Segmentation of the Shoulder from MRI with Inference Accuracy Prediction"

_diagnostics, 2023, doi:10.3390/diagnostics13101668_

Round 1
Reviewer 1 Report
This work presents Deep learning-based segmentation of the shoulder from MRI 2 with an inference accuracy prediction system, which has following observations that are to be addressed
The introduction part should include some recent works of literature, and its strengths and weaknesses.
Key contributions of this work should also be included in the introduction part
Novelty of this work also should be highlighted in this
Overall work should be presented as an architectural diagram
Figure 2 lacks clarity
The proposed algorithm in this work needs to be separately presented
Future work should be included
Unable to locate the comparison of results
Overall improvements are required from the introduction to the proposed methodology to its performance evaluation.
Minor editing would fix the issues
Author Response
Response to Reviewer #1
Ref.:
- Manuscript ID: diagnostics-2379685
- Title: Deep learning-based segmentation of the shoulder from MRI with inference accuracy prediction
The authors would like to thank you for the evaluation of our manuscript and for your considerate and helpful feedback. Please find below the point-by-point response to your comments:
-------------------------------------------------------------------------
This work presents Deep learning-based segmentation of the shoulder from MRI 2 with an inference accuracy prediction system, which has following observations that are to be addressed.
- The introduction part should include some recent works of literature, and its strengths and weaknesses.
Thank you for your comment, we agree that a more detailed summary of the related literature improves provides further context and highlights the novelty and impact of our work. The introduction has been extended and the recent relevant works are now described with a description of their strengths and weaknesses from line 54 in the manuscript:
“Medina et al. and Ro et al. reported the use of the U-net, a deep learning network architecture for semantic segmentation [1], to automatically segment the rotator cuff muscles on a single 2D slice (Y-view) of T1-weighted MRI for automatic fat fraction analysis [2][3]. In 3D, Godoy et al. reported the use of the U-net for automatic segmentation of the pectoralis major muscle on T1-weighted MRI to automatically evaluate its major cross sectional area. [4] Most recently, Riem et al. utilized the U-net for automatic segmentation and 3D fat fraction analysis of the rotator cuff muscles from clinical diagnostic sagittal T1-weighted MRI scans [5]. They reported mean and standard deviation segmentation accuracy of 0.92 ± 0.14 however, the method was trained and tested on data from only two centers and only segmentation of the sagittal MRI was performed, inhibiting the analysis of the muscle volume and the fat fraction in the entire muscle due to the restricted field of view.”
- Key contributions of this work should also be included in the introduction part. Novelty of this work also should be highlighted in this.
To highlight the novelty of our work we have augmented the introduction to include the following text (line 100). We believe that this, along with the extended context provided to previous work (see response 1) clarifies and highlights our contribution.
“To enable widespread accurate and efficient 3D measurement of RCT diagnostic factors we propose a fully automatic algorithm for 3D segmentation of the shoulder joint from diagnostic MRI with integrated automatic segmentation error checking. We present the use of the nnU-Net for shoulder anatomy segmentation from diagnostic MRI from all planes (axial, sagittal, transversal), verified on heterogenous data acquired in multiple centers. For efficient accuracy verification we present a study of automatic uncertainty-based label specific error checking, utilizing multiple segmentations generated from both the nnU-Net subnetworks and dropout during inference, integrated directly into the nnU-net framework.”
- Overall work should be presented as an architectural diagram.
As recommended by the reviewer we have included an architectural diagram of the overall work in order to improve clarity of the methodologies and contribution. Figure 1 (Line 177) has been augmented to provide an overview of our work with the help of an architectural diagram.
- Figure 2 lacks clarity
To improve the clarity of the usage of data within this study we have expanded Figure 2 (line 233) to show the distribution of the dataset for the different networks and tasks.
- The proposed algorithm in this work needs to be separately presented.
To ease the understanding of the final algorithm, Figure 1 (Line 177) has been augmented to include a diagram of the proposed solution. Contribution of this work (highlighted in blue) and its integration with the nnU-Net framework (highlighted in red) is depicted in Figure 1. In Figure 1 a) the calibration processes of the proposed algorithm are depicted, in Figure 1 b) the application of the final segmentation algorithm with error detection is shown and Figure c) shows how the performance of the algorithm was tested.
- Future work should be included.
Future work will be focussed in two primary areas: use of the algorithms for the analysis of multicenter clinical trials and, the error localisation and correction techniques. The following description of these activities has been added to the discussion section of the manuscript:
(1) clinical study data analysis for the evaluation of predictive factors for the outcome of rotator cuff repair (line 389):
“In the future, the proposed shoulder anatomy segmentation methods will be used as the basis for 3D analysis of the anatomy of rotator cuff patients. For example, in the multicentre study of arthroscopic rotator cuff repair [6], the algorithms are being applied to study the predictive value of 3D image based anatomical metrics such as whole muscle and regional fat fraction, muscle volume and 3D bone morphology metrics for repair outcome.“
(2) extension of the error detection methods to include automatic localisation of segmentation error and efficient correction for application in the routine clinical diagnosis and treatment decision making for rotator cuff tear patients. (line 371):
“Cases detected as insufficiently accurate would currently still require manual slice-by-slice verification and voxel-wise correction. In future work, pixel-wise or regional uncertainty metrics could be used to develop semiautomatic correction techniques [7]. Corrections based on the results of auxiliary networks, could be suggested to the user, reducing the need for voxel-wise correction.“
- Unable to locate the comparison of results:
Comparisons of our results are provided in sections 3.1, tables 2, 3 and 4 and figure 4. To improve clarity, we have ensured that all data is referenced in the text and descriptions have been augmented to improve clarity.
In addition, we have included a more detailed comparison of our results and previous results reported in the literature in two primary areas:
(1) the segmentation accuracies of the network proposed in this study compared to previous reports of similar segmentation networks (line 321):
“The accuracy of the automatic bone segmentation was found to be comparable to other reports of 3D automatic bone segmentation from MRI such as the use of a convolutional network for the segmentation of the pelvis and femur presented by Zeng et al. [8]. On the rotator cuff muscles, the segmentation network presented herein, when applied to multicenter data of all image planes, achieved similar segmentation accuracies to those reported by Riem et al. [5].”
(2) the performance of our proposed error detection methods compared to other similar reports within the literature (line 337)
“Overall, the coefficients of determination of the accuracy prediction metrics and the Dice coefficient presented herein, are lower compared to those presented by Roy et al. [9]. However, in this work, the values of the inference performance prediction metrics were compared to the Dice coefficient between the predicted and the manually corrected inference segmentation, rather than the Dice coefficient between predicted and original manual segmentation. Consequently, in this work, the maximal Dice coefficient of 1.0 represented any segmentation that was completely accepted by the expert without the need for correction was sometimes achieved. While we believe this measure is more representative of the state-of-the-art manual error verification process applied in clinical practice, it results in nonuniformly distributed Dice coefficients and lowers the coefficients of determination between the Dice and the presented inference performance predictors.”
Overall improvements are required from the introduction to the proposed methodology to its performance evaluation.
We thank the reviewer for the helpful and insightful feedback. Based on their questions and comments we have made significant changes to the manuscript, including to the introduction, methodologies and to the interpretation of results. We hope that these changes have improved the clarity and completeness of the manuscript and improved understanding of the novelty and impact or this presented work.
References
- Ronneberger, O.; Fischer, P.; Brox, T. U-Net: Convolutional Networks for Biomedical Image Segmentation, 2015. Available online: https://arxiv.org/pdf/1505.04597.
- Medina, G.; Buckless, C.G.; Thomasson, E.; Oh, L.S.; Torriani, M. Deep learning method for segmentation of rotator cuff muscles on MR images. Skeletal Radiol. 2021, 50, 683–692.
- Ro, K.; Kim, J.Y.; Park, H.; Cho, B.H.; Kim, I.Y.; Shim, S.B.; Choi, I.Y.; Yoo, J.C. Deep-learning framework and computer assisted fatty infiltration analysis for the supraspinatus muscle in MRI. Sci Rep 2021, 11, 15065.
- Godoy, I.R.B.; Silva, R.P.; Rodrigues, T.C.; Skaf, A.Y.; Castro Pochini, A. de; Yamada, A.F. Automatic MRI segmentation of pectoralis major muscle using deep learning. Sci Rep 2022, 12.
- Riem, L.; Feng, X.; Cousins, M.; DuCharme, O.; Leitch, E.B.; Werner, B.C.; Sheean, A.J.; Hart, J.; Antosh, I.J.; Blemker, S.S. A Deep Learning Algorithm for Automatic 3D Segmentation of Rotator Cuff Muscle and Fat from Clinical MRI Scans. Radiol. Artif. Intell. 2023, 5, e220132.
- Audigé, L.; Bucher, H.C.C.; Aghlmandi, S.; Stojanov, T.; Schwappach, D.; Hunziker, S.; Candrian, C.; Cunningham, G.; Durchholz, H.; Eid, K.; et al. Swiss-wide multicentre evaluation and prediction of core outcomes in arthroscopic rotator cuff repair: protocol for the ARCR_Pred cohort study. BMJ Open 2021, 11, e045702.
- Sander, J.; Vos, B.D. de; Išgum, I. Automatic segmentation with detection of local segmentation failures in cardiac MRI. Sci Rep 2020, 10.
- Zeng, G.; Schmaranzer, F.; Degonda, C.; Gerber, N.; Gerber, K.; Tannast, M.; Burger, J.; Siebenrock, K.A.; Zheng, G.; Lerch, T.D. MRI-based 3D models of the hip joint enables radiation-free computer-assisted planning of periacetabular osteotomy for treatment of hip dysplasia using deep learning for automatic segmentation. Eur. J. Radiol. Open 2021, 8, 100303.
- Roy, A.G.; Conjeti, S.; Navab, N.; Wachinger, C. Bayesian QuickNAT: Model uncertainty in deep whole-brain segmentation for structure-wise quality control. NeuroImage 2019, 195, 11–22.
Reviewer 2 Report
This manuscript presents a method based on the standard nnU-Net architecture for the segmentation of the structures of the shoulder from MR images.
While the results are plausible, the authors do not seem to share the data, nor the algorithm, nor the trained model, and independent reproduction of the results is therefore impossible.
The "finding" that nnU-Net is a valid architecture for medical image segmentation is by now overwhelmingly established, and does not bear any scientific interest.
In absence of the public release of the accompanying trained model, this paper brings no novel contribution to the current scientific literature, as it is merely describing a tool for internal use of the Authors and is of no use to the community at large.
Author Response
Response to Reviewer #2
Ref.:
- Manuscript ID: diagnostics-2379685
- Title: Deep learning-based segmentation of the shoulder from MRI with inference accuracy prediction
The authors would like to thank you for the evaluation of our manuscript and for your considerate feedback. Please find below the point-by-point response to your comments:
------------------------------------------------------------------------------------
This manuscript presents a method based on the standard nnU-Net architecture for the segmentation of the structures of the shoulder from MR images.
- While the results are plausible, the authors do not seem to share the data, nor the algorithm, nor the trained model, and independent reproduction of the results is therefore impossible.
Thank you for your comment. We agree that the value in AI based anatomy segmentation algorithms largely lies in their ability to facilitate further clinical research. For this reason, we have made our model for automatic segmentation of the shoulder in clinical MRI in combination with the error detection algorithm publicly available for non-commercial research purposes on request. For example, the presented model is currently being used in an external multicenter clinical trial for the development of a predictive model for rotator cuff outcome [1]. Additionally, to facilitate the re-evaluation of the proposed metrics and recalibration of the thresholds to be applied to other anatomical structures we have made the code of the augmented nnU-Net proposed in this study publicly available on GitHub (https://github.com/persmed/nnUNet_structure_wise_uncertainty).
To reflect this the data sharing statement has been updated and this information has also been included in the discussion section of the manuscript on line 387 and line 394:
” For use in furthers research, the code of the augmented nnU-Net used within this study has been made publicly available on github.”
“For use in future research the presented segmentation network with integrated error detection will be made available on request to the corresponding author.”
- The "finding" that nnU-Net is a valid architecture for medical image segmentation is by now overwhelmingly established and does not bear any scientific interest.
Yes, we agree that the nnU-Net is proved to perform extremely well on a high variety of datasets and that this framework is used by a large portion of the research community on their specific tasks. Despite this, a method for segmentation of the shoulder anatomy (using a U-net or otherwise) from MRI (acquired from a range of centres) required to facilitate a large amount of follow-up research pertaining to the 3D analysis of prognostic factors for rotator cuff repair, was yet to be reported. Also, even if models trained with the nnU-Net framework usually perform well during development, their performance during application is unknown and therefore their usage, particularly in larger clinical studies is greatly limited by the need for time consuming expert manual slice-by-slice verification. It is our believe and experience that this limitation is largely responsible for the very limited application of segmentation models in clinics and in research. To overcome this problem, we therefore augmented the widely used nnU-net framework to allow for automatic detection of inference segmentation error. Through the measurement of the segmentation variance based on predictions from the nnU-Net subnetworks, the proposed method provides valuable information without the need of expensive computational costs. It is expected that this method could also be used in other applications to facilitate wider use of segmentation models. The impact of such an error detection algorithm would be the elimination of the need to visually inspect every case while ensuring a high level of confidence in the use of the resulting 3D anatomical models for further clinical analysis. In our future work we aim to further reduce the user workload via automatic localisation of the error and guided corrections.
To highlight the novelty and impact of our work, we have augmented Figure 1 (Line 177) to provide an overview, with the help of an architectural diagram, to present our work (blue) in relation to the existing nnU-net framework (blue) We have additionally highlighted the impact and novelty of our work via significant changes to the introduction and discussion section of the manuscript.
- In absence of the public release of the accompanying trained model, this paper brings no novel contribution to the current scientific literature, as it is merely describing a tool for internal use of the Authors and is of no use to the community at large.
With the release of the trained model for automatic segmentation of the shoulder in multi-plane clinical MRI we provide a valuable tool for the research community on the shoulder. Modifications to the nnU-Net are being uploaded to GitHub for use by the wider public. With the release of the error detection algorithm, we also provide a tool to re-evaluate the presented algorithms and allow for recalibration of the thresholds for use in alternative task.
We thank the reviewer for the helpful and insightful feedback. Based on their questions and comments we have made significant changes to the manuscript, including to the introduction, methodologies and the interpretation of results. We hope that these changes have improved the clarity and completeness of the manuscript and improved understanding of our presented work.
References
- Audigé, L.; Bucher, H.C.C.; Aghlmandi, S.; Stojanov, T.; Schwappach, D.; Hunziker, S.; Candrian, C.; Cunningham, G.; Durchholz, H.; Eid, K.; et al. Swiss-wide multicentre evaluation and prediction of core outcomes in arthroscopic rotator cuff repair: protocol for the ARCR_Pred cohort study. BMJ Open 2021, 11, e045702.
Round 2
Reviewer 1 Report
Now the authors addressed the comments given and the presentation also improved substantially. So this manuscript can be accepted now
Reviewer 2 Report
Thank you for the changes to the manuscript. I understand that the statement "data available upon reasonable request" is a de facto standard, although it has turned out to be inefficient and often hindering the reproducibility of the work (see Tedersoo et al, 2021: https://doi.org/10.1038/s41597-021-00981-0). Usually, a proper license which includes the terms under which the work can be reused and how it should be properly cited should take care of any possible fears of unauthorized usage. If the authors are concerned about unauthorized commercialization, they can rest assured that no company would risk a lawsuit for this.
The effort of publishing the modified code is however commendable, and I do not wish to stand in the way of publishing this article.
So, all this will not impact the acceptance of the paper from my side, but as a reproducibility and open science advocate, I suggest the authors, if they don't wish to publicly release their model, at least add clear instructions to their github repository on how to obtain the model weights: i.e. who to contact, and what constitutes a "reasonable request," so that their work can reach the broadest audience. This will not only benefit the community, but the authors themselves, who will receive more citations and recognition.